

# Predicting danceability and song ratings using deep learning and auditory features

Wei Wu

Xiamen Medical College, Xiamen, Fujian, China

## ABSTRACT

Predicting a song's danceability and overall rating poses a significant challenge due to the complex interplay between musical characteristics and listener preferences. In this study, we propose a deep learning framework that jointly addresses the tasks of danceability estimation and popularity prediction. Our model integrates a Bidirectional Long Short-Term Memory (BiLSTM) network to capture sequential and contextual patterns from categorical inputs, alongside a Residual Network (ResNet) that extracts hierarchical representations from numerical auditory features. These complementary feature streams are fused using a cross-attention mechanism, enabling the model to effectively learn intricate relationships across heterogeneous data modalities. Experimental evaluations demonstrate that our approach consistently outperforms traditional machine learning baselines and recent deep learning models. The results demonstrate the effectiveness of cross-attention in structured music data modelling and highlight the framework's potential in advancing music recommendation and audio analysis systems.

## BACKGROUND AND MOTIVATION

The study of music popularity from the acoustic feature perspective has gained significant attention recently, particularly in the context of the so-called "Hit Song Science". This concept encapsulates the study of how various acoustic attributes of songs contribute to their commercial success and audience reception. While numerous studies have conducted into this area, the need for further research remains pressing, especially as the music industry evolves and the dynamics of listener preferences shift. The question of the contribution of acoustic features to the popularity of a song is fundamental, as it not only informs artists and producers of the elements that make a song likable to the audience but also helps in the development of predictive models that can identify the success of a song before release.

To understand the extent to which acoustic features determine the popularity of a song, an understanding of the specific features that contribute most significantly to this process is necessary. Previous research has identified a range of acoustic features, including tempo, loudness, energy, and danceability, as significant determinants of a song's popularity (*Sharma et al., 2022*; *North & Krause, 2024*). For instance, *Gao (2021)* studied streaming music popularity and utilized a dataset of over 130,000 tracks from Spotify, employing

Corresponding author
Wei Wu, xiamen2797@163.com

advanced machine learning models to classify songs based on their audio features and predict their popularity. This implies the existence of acoustic properties consistently linked to higher measures of popularity, offering a formulaic approach to creating hit songs. The role of cultural factors in shaping music preferences cannot be overlooked. The concept of cultural insularity, as discussed by *Demont-Heinrich (2020)*, illustrates how cultural consumers may exhibit inward tendencies in their music consumption, potentially limiting their exposure to diverse musical influences. This phenomenon underscores the importance of considering cultural contexts when analyzing music preferences and popularity. Additionally, the work of *Klarlund et al. (2023)* on perceived cultural distance in music perception highlights how cultural backgrounds can influence listeners' responses to music, further complicating the relationship between audio features and consumer preferences.

Danceability, as a musical concept, refers to how suitable a song is for dancing, and is typically quantified based on a combination of tempo, rhythmic stability, beat strength, and overall regularity in the acoustic signal. It captures the extent to which a track invites physical movement, making it a crucial dimension of listener interaction, particularly in popular music genres. As a feature derived from audio analysis, danceability has received considerable attention in recent studies for its role in shaping audience engagement and commercial performance. In their study, *Melchiorre & Schedl (2020)* demonstrated that danceability alongside energy and valence was significantly correlated with streaming metrics such as play counts and playlist inclusion, suggesting that highly danceable tracks are more likely to reach mainstream audiences and perform well on platforms like Spotify. This aligns with findings from *Su et al. (2022)*, who identified danceability as one of the key audio features contributing to the virality of music content on social media platforms, including TikTok and Spotify. Their work highlights how danceability not only reflects musical structure but also shapes listener behavior in algorithm-driven environments.

The evolution of music preferences in the digital age has also been influenced by technological advancements that facilitate music production and distribution. The integration of machine learning and artificial intelligence in music analysis has provided new insights into consumer behavior and preferences. For instance, *Xie (2024)* discusses how deep learning algorithms have transformed the understanding of consumer preferences in folk music, highlighting the role of technology in shaping contemporary music consumption. This technological shift aligns with the findings of *Raza & Nanath (2020)*, who noted that some studies found no strong correlation between audio features and popularity, suggesting that the complexities of consumer preferences extend beyond mere auditory characteristics. For example, the work of *Lee & Lee (2018)* emphasizes the importance of analyzing various popularity metrics to understand the characteristics that define successful music. By leveraging machine learning algorithms, researchers can not only classify songs but also predict their potential for popularity based on these acoustic features.

The research objective of this study is particularly relevant in the context of the music market, which is projected to experience a revenue growth rate of 10.03% by *Sharma et al. (2022)*. This growth underscores the importance of understanding the factors that drive

music consumption and popularity. By employing machine learning algorithms, both classification and regression techniques can be utilized to predict a song's popularity based on its acoustic features, thereby providing valuable insights for artists, producers, and marketers alike. The integration of data from the Spotify application programming interface (API) further enriches this analysis, as it offers a wealth of information regarding song characteristics and listener engagement across different geographical regions (*Scarratt et al., 2021*).

Song rating and danceability prediction remains a complex task involving understanding numerous musical features, user tastes, and context information. The prevailing machine learning methods are likely to rely on hand-crafted features or shallow feedforward neural networks, which cannot capture intricate interdependencies between musical features. The models also fail to effectively process heterogeneous data types, limiting their prediction. To address these limitations, we propose a hybrid deep learning architecture that combines Bidirectional Long Short-Term Memory (BiLSTM) (*Schuster & Paliwal, 1997*) based encoding for categorical features with a Residual Network (*He et al., 2015*) for numerical features. The LSTM block enables effective learning of sequential and contextual relationships in categorical data, while the Residual Network ensures stable feature extraction from numerical inputs. However, separately processing these feature types is insufficient, as real-world song ratings are influenced by complex interactions between musical metadata and numerical attributes such as tempo, loudness, and spectral features. To bridge this gap, we employ a cross-attention mechanism that integrates outputs from both the BiLSTM and Residual Network. Cross-attention enables the model to dynamically learn dependencies between categorical and numerical features, enhancing its ability to make accurate predictions. This method not only improves representation learning but also reduces information loss, making it particularly effective in handling the intricate relationships present in song rating data. Furthermore, our proposed architecture improves computational efficiency by leveraging structured processing pathways for different data modalities. This approach ensures that both categorical and numerical features contribute optimally to the prediction process, leading to better generalization and robustness across diverse datasets. By adopting this novel fusion strategy, we aim to push the boundaries of song rating prediction, enhancing both interpretability and predictive accuracy.

## RELATED WORK

Traditional approaches to predicting song popularity have evolved over the decades, primarily focusing on statistical analysis and early computational methods. In 2006, *Salganik, Dodds & Watts (2006)* investigated how social influence shapes song popularity and found that while song quality played a role, social influence greatly contributed to making certain songs become hits. By 2009, peer-to-peer (P2P) network data were used by *Koenigstein, Shavitt & Zilberman (2009)* to predict Billboard rankings and demonstrated that early trends in downloads could be predictive of future success. In 2010, *Bertin-Mahieux et al. (2008)* applied machine learning to the Million Song Dataset for automatic song classification based on acoustic features, although issues of feature selection and

overfitting remained. In 2011, *Pachet & Roy (2011)* expressed skepticism about the feasibility of hit song prediction, arguing that current models lacked sufficient accuracy to be considered a true science. However, in 2012, *Ni et al. (2012)* introduced the shifting perceptron algorithm, incorporating novel audio features to improve prediction accuracy, presenting a more optimistic outlook on the potential of hit song prediction. These researches determine the move away from traditional statistical tests towards machine learning algorithms and both emphasize the promise and the challenges of the field.

The advent of machine learning and deep learning has revolutionized the field of song popularity prediction. In 2017, *Yang et al. (2017)* employed convolutional neural networks (CNNs) in order to investigate audio features from songs directly, a significant break from previous methods that relied heavily on engineered features. This process demonstrated that it was possible for deep learning to discover intricate patterns in audio information that were previously not taken into consideration. By 2020, multimodal data source fusion became the focus of research attention. For example, *Martin-Gutierrez et al. (2020)* introduced a new end-to-end deep learning model that fused audio features with social media metrics to enhance prediction performance. This multimodal approach recognized the importance of contextual data, such as user engagement and trends on platforms like Spotify and YouTube, in predicting a song's success. In 2021, the use of advanced machine learning models, including random forests and support vector machines, gained traction. These models were applied to large datasets comprising audio features, lyrics, and metadata, allowing for a more nuanced understanding of the factors influencing song popularity (*Sharma et al., 2022*; *Gao, 2021*). The findings indicated that random forests consistently outperformed other models in terms of accuracy and interpretability, suggesting a robust framework for future research. In 2023, *Zhao et al. (2023)* field witnessed further advancements with the introduction of engineered metadata features that combined traditional acoustic analysis with modern machine learning techniques. Studies demonstrated that incorporating features such as lyrical sentiment and genre-specific characteristics significantly improved prediction outcomes, highlighting the necessity of a holistic approach to song popularity prediction.

## DATASET AND PREPROCESSING

### Data collection

The dataset used in this project originates from Kaggle and contains approximately 19,000 songs with 15 features. Each observation represents an individual song. The majority of these features are numerical values that quantify various musical attributes such as loudness, instrumentalness, danceability, and liveness. Some columns consist of string-based data, a few of which were dropped, while others were converted from numeric to categorical values to better represent their information. The dataset includes two target columns: **song_popularity**, which assigns a popularity score from 0 to 100 for the popularity prediction task, and **danceability** for the danceability prediction task. The key features of the dataset are summarized in Table 1.

**Table 1 Feature descriptions of the dataset.**

| Feature | Description |
|---|---|
| duration_ms | The duration of the track in milliseconds. |
| Key | The estimated key of the track, represented as an integer using Pitch Class notation (*e.g.*, 0 = C, 1 = C#/D♭, 2 = D, *etc.*). If no key is detected, the value is −1. |
| audio_mode | The modality of the track, where 1 represents major and 0 represents minor. |
| time_signature | The estimated time signature of the track, indicating the number of beats per measure. |
| Acousticness | A confidence score (0.0 to 1.0) indicating the likelihood that the track is acoustic, with 1.0 representing high confidence. |
| Energy | A score (0.0 to 1.0) reflecting the intensity and activity of the track, where high-energy tracks tend to be loud and fast-paced. |
| Instrumentalness | A score (0.0 to 1.0) estimating the likelihood that a track contains no vocals, where higher values indicate a higher probability of being an instrumental track. |
| Loudness | The overall loudness of the track in decibels (dB), averaged across its duration. Values typically range between −60 dB and 0 dB. |
| Speechiness | A score (0.0 to 1.0) measuring the presence of spoken words in a track. Higher values indicate more speech-like content, with values above 0.66 representing primarily spoken content. |
| audio_valence | A score (0.0 to 1.0) representing the musical positiveness conveyed by a track. High valence suggests positive emotions (*e.g.*, happy, cheerful), while low valence indicates negative emotions (*e.g.*, sad, angry). |
| Tempo | The overall estimated tempo of a track in beats per minute (BPM), indicating its speed or pace. |
| Liveness | A score (0.0 to 1.0) estimating the presence of an audience in the recording, where higher values suggest a higher probability that the track was performed live. |
| **song_popularity** (label) | A score (0–100) indicating the popularity of a song based on Spotify audience ratings. |
| **Danceability** (label) | A measure (0.0 to 1.0) of how suitable the track is for dancing, based on factors like tempo, rhythm stability, and beat strength. |

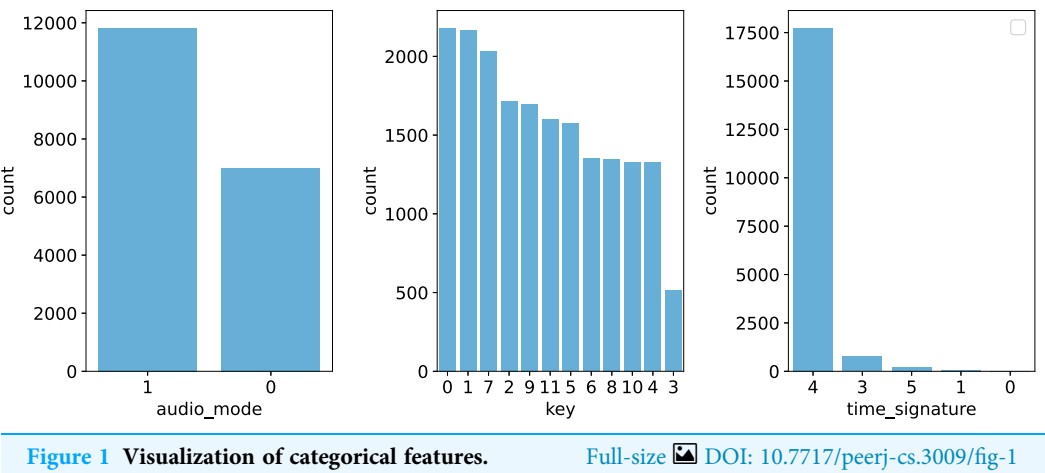

**Figure 1 Visualization of categorical features.**

To better illustrate the distribution of features, Figs. 1 and 2 provide graphical representations of categorical and numerical features, respectively. Figure 3 visualizes box plots of song_popularity and danceability in the dataset. These visualizations offer insights into the structure and variability of the dataset.

## Data preprocessing

To ensure the dataset's compatibility and enhance the performance of the predictive model, several preprocessing steps were applied. The dataset consists of approximately

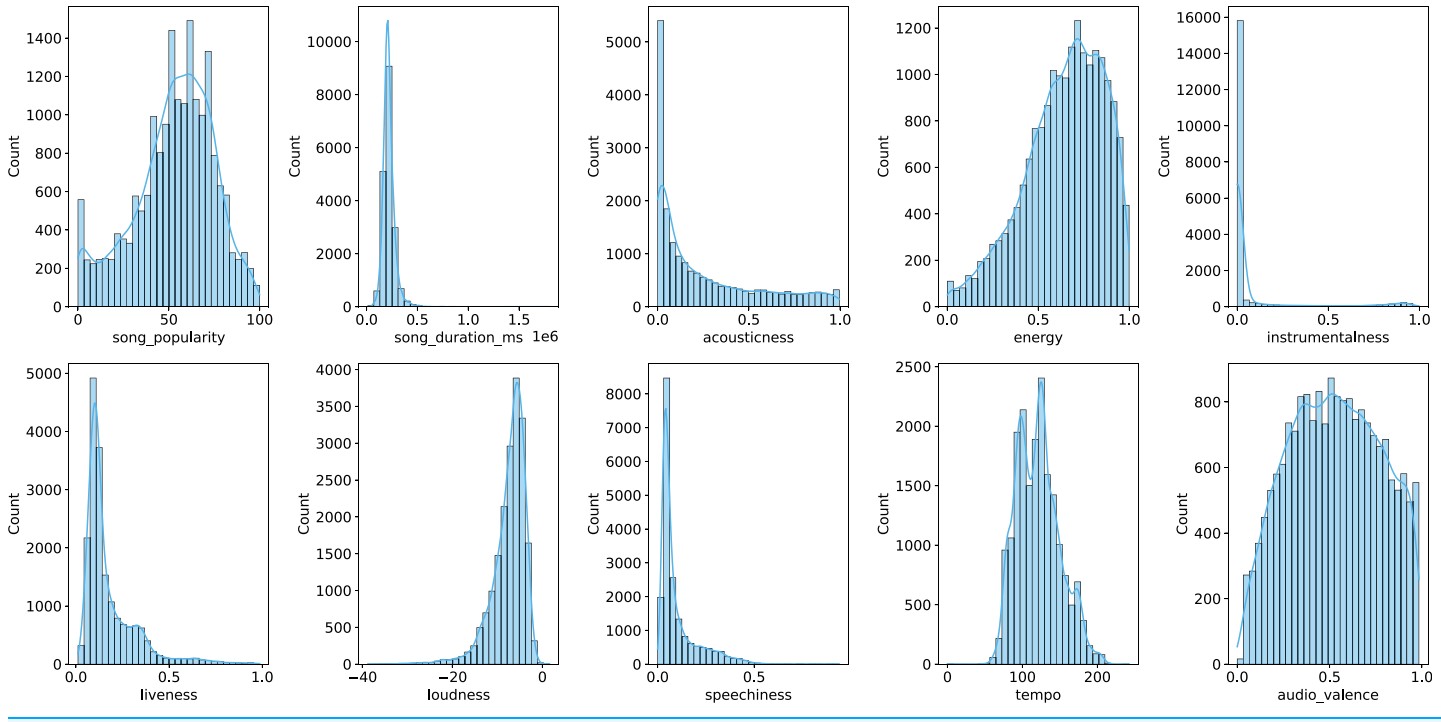

**Figure 2** Visualization of numerical features.

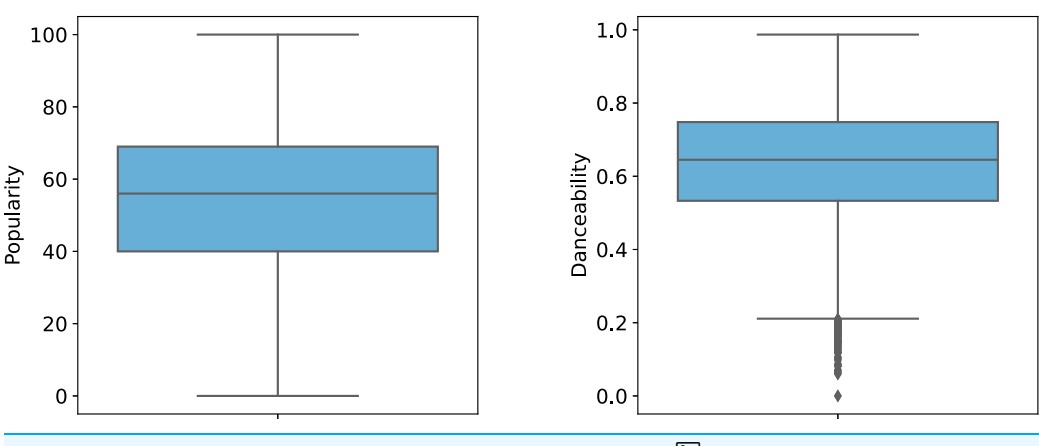

**Figure 3** Box plot of song popularity and danceability.

19,000 songs with 15 features, where most attributes are numerical, quantifying musical characteristics such as loudness, instrumentalness, danceability, and liveness. Some features were categorical, requiring encoding to facilitate their use in machine learning algorithms.

First, the dataset was loaded from a CSV file using Pandas. Features were classified into categorical (audio_mode, key, time_signature) and numerical types, with song_name excluded due to its irrelevance to the prediction task. Label encoding was applied to categorical features, converting them into numerical representations while preserving their distinctiveness. Subsequently, numerical features were standardized using the

**Table 2 Statistics of data samples in training, validation, and test sets.**

| Dataset split | Number of samples |
| --- | --- |
| Training set | 15,068 |
| Validation set | 1,884 |
| Test set | 1,883 |
| **Total** | **18,835** |

StandardScaler to ensure uniform scaling, which stabilizes the learning process. For the regression task, the label, which originally ranged from 0 to 100, was normalized to the range [0,1]. This transformation ensures that all values lie within a fixed, interpretable scale, preventing potential numerical instability in gradient-based optimization. Additionally, normalizing the target variable can help models converge faster by reducing the impact of different magnitudes between input features and the label. Following preprocessing, the dataset was randomly split into training, validation, and test sets in an 80-10-10 ratio. The test set was separated first, followed by an additional split of the training set to create the validation data. This approach ensures effective model evaluation and minimizes the risk of overfitting. The statistical information of the data in each of the training, validation, and test sets is summarized in the Table 2. These preprocessing steps collectively optimize the dataset, enabling the development of robust predictive models for estimating song popularity based on musical attributes.

## METHODOLOGY

### Selection method

The selection of modelling techniques for this study was driven by the nature of the input data and the dual prediction task involving both danceability and song rating. The dataset contains heterogeneous features, including categorical metadata (*e.g.*, genre, artist) and numerical auditory descriptors (*e.g.*, tempo, loudness, spectral features), each requiring specialised handling. To effectively model the temporal and contextual patterns in the categorical features, we selected a BiLSTM network, which is well-suited for capturing sequential dependencies in structured inputs. For the numerical auditory features, we employed a Residual Network (ResNet), chosen for its ability to extract deep hierarchical representations while preserving gradient flow through residual connections.

Given the complementary nature of these feature types, we adopted a cross-attention mechanism to integrate the BiLSTM and ResNet outputs. This allows the model to dynamically learn complex interdependencies between the two modalities, enhancing its capacity to understand how auditory signals and metadata jointly influence a song's perceived danceability and popularity. The selection of this hybrid framework was also informed by preliminary experiments, where simpler concatenation approaches yielded lower performance and failed to fully capture cross-modal relationships. Therefore, the final architecture was chosen based on both theoretical suitability and empirical validation.

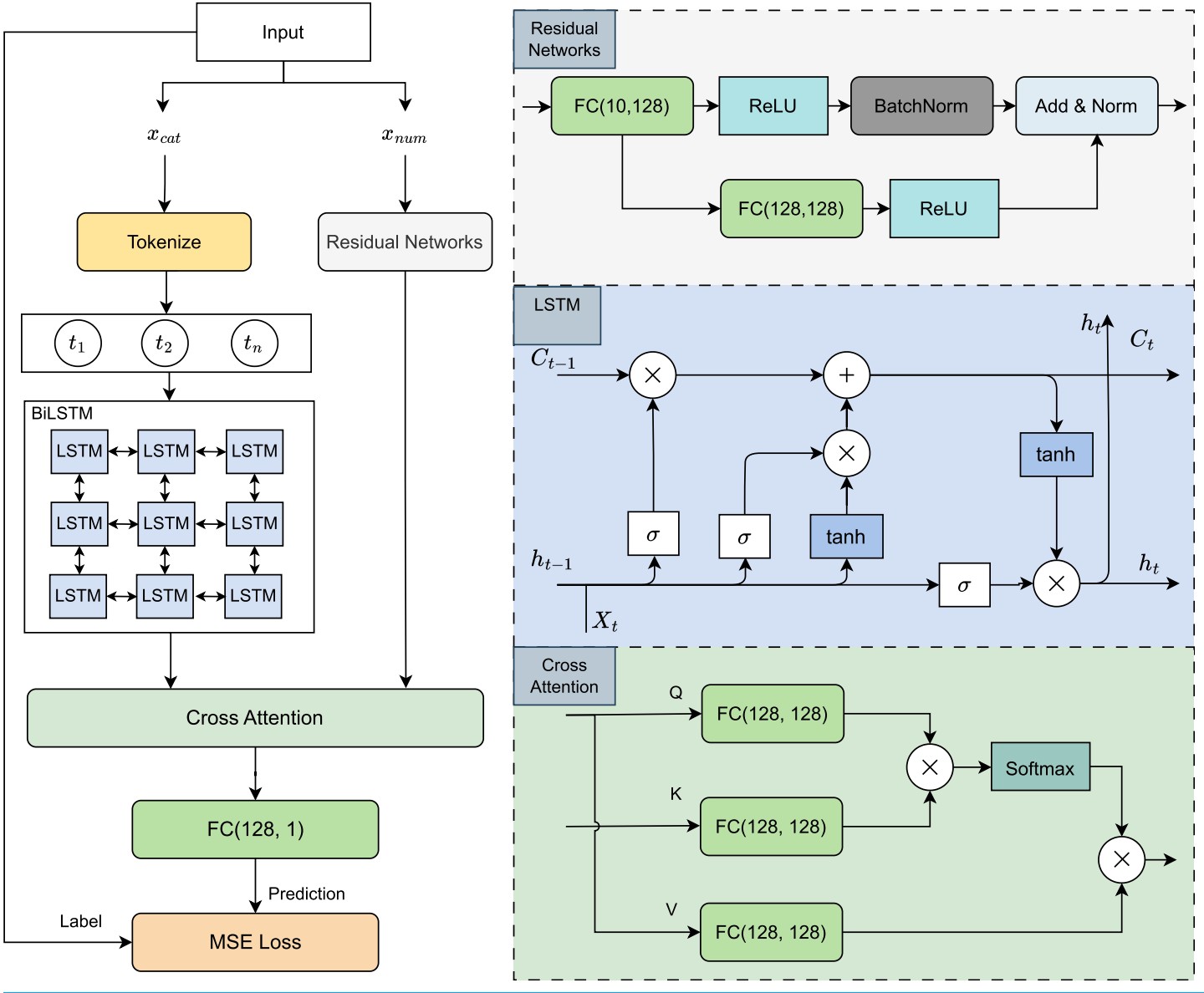

**Figure 4  Model architecture visualization.**

## Model architecture

Building on the selection rationale outlined above, we designed a hybrid deep learning architecture that leverages the strengths of both sequence modelling and hierarchical feature extraction. As illustrated in Fig. 4, the proposed model consists of two parallel processing streams tailored to the distinct modalities in the dataset.

Categorical features, such as genre or artist, are first embedded and passed through a BiLSTM network to capture temporal and contextual dependencies across sequences. In parallel, numerical auditory features, including tempo, spectral characteristics, and rhythmic patterns, are processed by a ResNet module to learn deep, multi-level representations that retain important low- and high-level acoustic information.

The outputs from both branches are then integrated using a cross-attention mechanism, which enables the model to dynamically align and weigh relevant information across the two modalities. This fusion strategy allows for the discovery of intricate relationships between auditory signals and categorical context. The combined representation is subsequently passed through a fully connected layer, which generates the final predictions for both danceability and overall song rating.

This architecture is designed to exploit the complementary strengths of BiLSTM and ResNet while enabling cross-modal interaction through attention, making it well-suited for modelling the complex dynamics of music perception and listener preferences.

### Song representation

The input dataset consists of categorical features, denoted as $x_{cat}$, and numerical features, denoted as $x_{num}$. These two feature sets are processed separately using two distinct neural network modules: a BiLSTM block for categorical data and a ResNet-based architecture for numerical data.

Categorical features $x_{cat}$ are processed using a BiLSTM-based model, which captures temporal and contextual dependencies between categorical variables through recurrent connections. The BiLSTM block processes the sequence of embedded categorical features as follows:

$$h_t, c_t = \text{LSTM}(x_t, h_{t-1}, c_{t-1}) \tag{1}$$

where:

- $x_t$ represents the input embedding at time step $t$,
- $h_t$ and $c_t$ are the hidden and cell states at time step $t$.

Numerical features $x_{num}$ are processed using a Residual Network (ResNet), which is designed to efficiently learn deep feature representations while maintaining gradient stability. The ResNet block consists of a main branch and a residual branch. The main branch includes a sequence of fully connected layers (FC (10, 128)), Rectified Linear Unit (ReLU) activation function and Batch Normalization (BatchNorm). The residual branch consists fully connected layers (FC (10, 128)) and ReLU activation function, ensuring that the depth of the network does not degrade performance. The final output from both branches is combined through normalized addition (Add & Norm).

### Feature fusion via cross-attention

To effectively integrate the extracted features from categorical and numerical inputs, we utilize a cross-attention mechanism. Let the output of the BiLSTM block be denoted as $T_{out}$ and the output of the Residual Network be denoted as $R_{out}$. In the cross-attention mechanism, $T_{out}$ serves as the query matrix $Q$, while $R_{out}$ provides the key-value pairs $(K, V)$. The attention scores are computed using the dot-product attention formula:

$$\text{Attention Scores} = \frac{QK^T}{\sqrt{d_k}}, \tag{2}$$

where

- $Q$ is the query matrix derived from $T_{out}$, with shape $(n_q, d_k)$,
- $K$ is the key matrix derived from $R_{out}$, with shape $(n_k, d_k)$.

The attention weights are obtained by applying a softmax function to the attention scores:

$$\text{Attention Weights} = \text{Softmax}\left(\frac{QK^T}{\sqrt{d_k}}\right). \tag{3}$$

This normalization ensures that attention weights are between 0 and 1 and sum to 1. The final output of the cross-attention mechanism is computed as:

$$\text{Output} = \text{Attention Weights} \times V, \tag{4}$$

where $V$ is the value matrix obtained from $R_{out}$, with shape $(n_k, d_v)$. The fused feature representation is passed through a fully connected layer with dimensions (128,1). This final layer produces the predicted song rating, enabling accurate assessment of song quality based on extracted features.

## Model configuration and training strategy

To support replicability and provide transparency in architectural design choices, we detail both the model configuration and training strategy employed in this study.

### Model configuration

The architecture was configured to balance model expressiveness and training efficiency. For the categorical input branch, each categorical variable was first embedded into a dense vector space with an embedding dimension of 32. These embeddings were concatenated and passed through a BiLSTM layer with a hidden dimension of 64 units per direction (total 128). To reduce overfitting, a dropout layer with a rate of 0.3 was applied after the BiLSTM output. For the numerical input branch, a custom Residual Network (ResNet) was designed. The main path consisted of two fully connected layers: FC(10, 128) followed by ReLU activation and batch normalization. The residual path included a parallel FC(10, 128) with ReLU activation. The outputs of both paths were combined using element-wise addition followed by layer normalization. The cross-attention fusion module uses attention dimension $d_k$ and $d_v$ were both set to 128. The result of the attention mechanism was passed through a fully connected layer with output dimension 128, followed by a final prediction layer FC(128, 1) for each task. All layers were initialized using Xavier uniform initialization, and layer normalization was applied where appropriate.

### Training strategy

The model was trained using the Adam optimizer with an initial learning rate of $1 \times 10^{-3}$. Training was performed over 50 epochs with a batch size of 128, balancing computational efficiency and convergence stability. To further mitigate overfitting, early stopping was employed with a patience of seven epochs, and dropout regularization was applied in the

fully connected layers. This training configuration yielded stable convergence and robust performance across multiple runs.

## EXPERIMENTAL SETUP

### Computing resource

All experiments were conducted on a high-performance computing environment equipped with an Intel Xeon processor, 32 GB of RAM, and an NVIDIA RTX 3070 GPU with 12 GB of VRAM. The models were implemented using Python with the PyTorch framework. Data preprocessing and analysis were performed using Pandas and NumPy, while visualization was conducted with Matplotlib and Seaborn. The deep learning training process leveraged CUDA optimizations to enhance computational efficiency. Hyperparameter tuning was carried out to ensure optimal model performance. The entire pipeline was executed on Ubuntu 20.04 to provide a stable and efficient computational environment.

### Evaluation metrics

To assess the performance of our model, we employ four widely used evaluation metrics: mean squared error (MSE), the coefficient of determination ($R^2$), mean absolute error (MAE), and root mean square deviation (RMSE). These metrics provide a robust and comprehensive analysis of the model's predictive capabilities.

- **Mean squared error**: measures the average squared difference between actual and predicted values, indicating the overall prediction error:

$$\text{MSE} = \frac{1}{n} \sum_{i=1}^{n} (y_i - \hat{y}_i)^2. \tag{5}$$

- **Coefficient of determination**: represents the proportion of variance in the dependent variable explained by the model, with higher values indicating better performance:

$$R^2 = 1 - \frac{\sum_{i=1}^{n} (y_i - \hat{y}_i)^2}{\sum_{i=1}^{n} (y_i - \bar{y})^2}. \tag{6}$$

- **Mean absolute error**: computes the average absolute difference between actual and predicted values, offering an interpretable error measure:

$$\text{MAE} = \frac{1}{n} \sum_{i=1}^{n} |y_i - \hat{y}_i|. \tag{7}$$

- **Root mean square deviation**: calculates the square root of MSE, providing a more intuitive measure of error by maintaining the same unit as the target variable:

$$\text{RMSE} = \sqrt{\frac{1}{n} \sum_{i=1}^{n} (y_i - \hat{y}_i)^2}. \tag{8}$$

These metrics collectively ensure a balanced evaluation of model accuracy, variance explanation, and robustness, allowing a thorough comparison with baseline and state-of-the-art models.

**Table 3 List of tuned hyperparameter in machine learning models.**

| Model | Hyperparams | Range of value |
|---|---|---|
| *k*-Nearest Neighbors (*k*-NN) | Number of neighbors | 3, 5, 7, 9 |
| AdaBoost | Number of estimators | 100, 200, 300 |
| Decision Tree | Max depth | 3, 5, 7 |
| Random Forest | Number of estimators | 100, 200, 300 |
| | Max features | 1, 10, 'log2', 'sqrt' |
| XGBoost | Number of estimators | 100, 200, 300 |
| | Learning rate | 0.01, 0.1, 0.2 |

## Evaluation strategy

### Machine learning models

In this section, we evaluate the performance of the proposed model and compare it with several widely used machine learning methods. Specifically, the models employed in this study include Random Forest (*Breiman, 2001*), K-Nearest Neighbors (*Kramer, 2013*), Decision Tree (*Rokach & Maimon, 2005*), AdaBoost (*Beja-Battais, 2023*), and XGBoost (*Chen & Guestrin, 2016*). Each of these models has been selected due to its strong performance in regression tasks and its ability to capture different aspects of the dataset's underlying structure. These models are trained and evaluated using multiple performance metrics to assess their ability to predict song ratings and danceability accurately. The parameters of each model were optimized through hyperparameter tuning to ensure a fair comparison. The tuning process for each model is summarized in Table 3.

### Deep learning models

In addition, we conduct comparing performance of our model with several state-of-the-art deep learning models designed for tabular data including:

- **Multi-Layer Perceptron (MLP)** (*Bourlard & Morgan, 1994*) serves as a baseline deep learning approach, where fully connected layers process numerical and categorical features. While MLP can capture complex patterns, it lacks explicit mechanisms for feature selection and interaction modeling, which are crucial for tabular data.
- **Deep & Cross Network (DCN)** (*Wang et al., 2021*) enhances MLP by introducing explicit feature crossing at different levels, improving its capability to capture non-linear interactions efficiently. This structured feature interaction modeling can provide an advantage in learning song rating patterns.
- **TabNet** (*Arik & Pfister, 2021*) utilizes a sequential attention mechanism to select relevant features dynamically, enabling interpretability while maintaining high predictive accuracy. Its ability to model feature interactions adaptively makes it well-suited for tabular data tasks.
- **TabTransformer** (*Huang et al., 2020*) leverages a Transformer-based architecture to model dependencies between categorical and numerical features. By applying self-attention mechanisms, it effectively captures feature relationships, improving generalization in tabular learning tasks.

- **FT-Transformer** (*Gorishniy et al., 2021*) revisits Transformer architectures for tabular data by combining feature tokenization and column embedding strategies. It employs a simple yet effective design where each feature is treated as a token, enabling the self-attention mechanism to model complex feature interactions.
- **Graph Neural Machine (GNM)** (*Nikolentzos et al., 2024*) represents tabular data as a graph and utilizes message-passing mechanisms to enhance feature interactions. By leveraging the structural relationships between features and samples, GNM can improve learning performance, especially in complex rating prediction scenarios.

Our comparative analysis highlights the strengths and limitations of these models, demonstrating how our proposed approach achieves competitive or superior performance while maintaining interpretability and efficiency in song rating prediction.

## RESULTS AND DISCUSSION

### Ablation study

To further investigate the importance of individual components within our proposed architecture, we conducted a series of ablation experiments. These experiments were designed to systematically remove or replace key elements of the model, thereby isolating their effects on the prediction performance. The following ablation variants were implemented and evaluated:

- **Without ResNet:** the ResNet module responsible for processing numerical features was removed. Numerical inputs were instead passed through a single fully connected layer without residual connections, allowing us to evaluate the role of deep residual learning in capturing complex numerical patterns.
- **Without BiLSTM:** the BiLSTM block used for capturing contextual dependencies in categorical features was excluded. Categorical data were encoded and passed through a shallow dense layer, testing the importance of sequential modeling in this context.
- **Replacing Cross-Attention with Concatenation:** instead of using a cross-attention mechanism to fuse the BiLSTM and ResNet outputs, we concatenated both feature vectors and passed them directly to the final prediction layer. This setup explores whether attention-based integration is superior to simple feature merging.
- **Replacing BiLSTM with GRU** (*Cho et al., 2014*): to assess the sensitivity of the model to the choice of sequence encoder, we replaced the BiLSTM with a Gated Recurrent Unit (GRU). This variant maintains the same embedding and attention pipeline, providing insight into alternative temporal modeling strategies.
- **Replacing BiLSTM with Transformer Encoder** (*Vaswani et al., 2017*): in this configuration, we replaced the BiLSTM with a Transformer encoder to leverage its ability to model long-range dependencies and capture global context through self-attention mechanisms. Additionally, the Transformer enables more efficient parallel processing, leading to faster training and improved performance on large datasets.

**Table 4 Performance of the model with different settings.**

| Model | Popularity task | | | | Danceability task | | | |
|---|---|---|---|---|---|---|---|---|
| | MAE | MSE | RMSE | R2 | MAE | MSE | RMSE | R2 |
| Without ResNet | 0.1395 | 0.0361 | 0.1899 | 0.2810 | 0.0721 | 0.0104 | 0.1020 | 0.5976 |
| Without BiLSTM | 0.1412 | 0.0373 | 0.1932 | 0.2604 | 0.0735 | 0.0109 | 0.1044 | 0.5812 |
| Without cross-attention | 0.1367 | 0.0347 | 0.1862 | 0.2982 | 0.0704 | 0.0098 | 0.0990 | 0.6147 |
| BiLSTM → GRU | 0.1309 | 0.0327 | 0.1809 | 0.3421 | 0.0680 | 0.0093 | 0.0964 | 0.6289 |
| BiLSTM → Transformer | 0.1321 | 0.0331 | 0.1819 | 0.3367 | 0.0689 | 0.0095 | 0.0974 | 0.6240 |
| Ours (Full model) | **0.1284** | **0.0315** | **0.1774** | **0.3698** | **0.0667** | **0.0089** | **0.0942** | **0.6443** |

Note:
Bold denotes the best-performing value for each metric.

The ablation results in Table 4 highlight the importance of each architectural component in our proposed model. Removing the ResNet block leads to a noticeable degradation in both tasks, particularly in the popularity prediction ($R^2$ drops from 0.3698 to 0.2810) and danceability prediction ($R^2$ drops from 0.6443 to 0.5976), indicating its critical role in capturing informative visual features. Similarly, excluding the BiLSTM module causes a reduction in performance across all metrics, reaffirming its effectiveness in modeling temporal dependencies. The cross-attention mechanism also proves essential, as its removal leads to a consistent performance drop in both tasks. For instance, $R^2$ on the danceability task falls from 0.6443 to 0.6147, demonstrating the advantage of guided interaction between multimodal features over naive concatenation. Additionally, replacing BiLSTM with alternative sequence encoders, both GRU and Transformer variants yield slightly inferior results. Additionally, replacing BiLSTM with an alternative sequence encoder, bib8, yields a slightly better result. The GRU-based model achieves an $R^2$ of 0.3421 (popularity) and 0.6289 (danceability). The GRU-based model achieves an $R^2$ of 0.3421 (popularity) and 0.6289 (danceability), while the Transformer-based version records 0.3367 and 0.6240, respectively. These results suggest that although the architecture is relatively robust to changes in the sequence encoder, BiLSTM remains the most effective choice for our tasks. Overall, the full model outperforms all ablated versions, validating the design choices made in our framework.

## Comparative study

The results presented in Table 5 confirm the effectiveness of the proposed model in both prediction tasks. Our model achieves the best performance across all evaluation metrics, with the lowest MAE (0.1228 and 0.0667), MSE (0.0297 and 0.0089), and RMSE (0.1723 and 0.0942) for the popularity and danceability tasks, respectively. Additionally, it attains the highest $R^2$ scores—0.3842 for popularity and 0.6443 for danceability—indicating that it captures a substantially greater portion of variance compared to all baseline models. Among the traditional machine learning methods, XGBoost yields the best performance, achieving an MAE of 0.1613 (popularity) and 0.0906 (danceability), with corresponding $R^2$ scores of 0.1407 and 0.4831. This highlights the effectiveness of gradient boosting in handling structured features. Random Forest and AdaBoost perform comparably,

**Table 5 Performance comparison with machine learning models on the test set.**

| Model | Popularity task | | | | Danceability task | | | |
|---|---|---|---|---|---|---|---|---|
| | MAE | MSE | RMSE | R2 | MAE | MSE | RMSE | R2 |
| Random Forest | 0.1638 | 0.0426 | 0.2065 | 0.1155 | 0.0854 | 0.0117 | 0.1082 | 0.5305 |
| k-Nearest Neighbors | 0.1674 | 0.0445 | 0.2109 | 0.0774 | 0.1020 | 0.0161 | 0.1267 | 0.3563 |
| Decision Tree | 0.1658 | 0.0437 | 0.2090 | 0.0942 | 0.0969 | 0.0146 | 0.1209 | 0.4140 |
| AdaBoost | 0.1637 | 0.0426 | 0.2065 | 0.1157 | 0.0935 | 0.0137 | 0.1169 | 0.4518 |
| XGBoost | 0.1613 | 0.0414 | 0.2035 | 0.1407 | 0.0906 | 0.0129 | 0.1135 | 0.4831 |
| Ours | **0.1228** | **0.0297** | **0.1723** | **0.3842** | **0.0667** | **0.0089** | **0.0942** | **0.6443** |

**Note:**
Bold denotes the best-performing value for each metric.

benefiting from ensemble learning, though their predictive power remains clearly lower than the proposed model. On the other hand, Decision Tree and k-Nearest Neighbors exhibit weaker performance, likely due to their limited generalization capabilities and tendency to overfit or underfit complex patterns in the data.

In summary, the experimental results validate the effectiveness of our approach, demonstrating a substantial improvement over traditional methods. The performance of the proposed model can be attributed to its ability to capture complex relationships within the dataset, possibly leveraging deep learning or more advanced ensemble techniques.

The performance comparison in Table 6 highlights the effectiveness of our proposed model over a range of state-of-the-art deep learning approaches tailored for tabular data. Across both prediction tasks, our model consistently achieves the best results in all evaluation metrics. Specifically, it records the lowest MAE (0.1228 for popularity and 0.0667 for danceability), MSE (0.0297 and 0.0089), and RMSE (0.1723 and 0.0942), demonstrating both high accuracy and reduced prediction error. Moreover, it attains the highest $R^2$ scores (0.3842 and 0.6443), indicating a substantially better ability to explain the variance in the target variables compared to all baseline deep learning models. Among the compared methods, GNM shows the strongest baseline performance, achieving an $R^2$ of 0.2219 (popularity) and 0.5737 (danceability). However, our model outperforms GNM by a notable margin, particularly in the popularity task, where the $R^2$ improvement exceeds 16 percentage points. Additionally, both TabTransformer and FT-Transformer deliver competitive results, reflecting the strength of attention mechanisms in capturing complex feature interactions. FT-Transformer improves upon TabTransformer with a lower MAE (0.1428 *vs.* 0.1541) and higher $R^2$ (0.2450 *vs.* 0.2130) in the popularity task, but still falls short of our model's performance across all metrics. These gains suggest that our model captures more complex and relevant relationships within the data, potentially due to improved architectural design or better multimodal feature integration. While MLP, DCN, and TabNet provide moderate results, they consistently lag behind in both tasks, underscoring the limitations of standard or shallow deep learning architectures when applied to this problem domain. In contrast, our model's strong performance across the board indicates enhanced representational capacity and generalization ability.

**Table 6  Performance comparison of tabular deep learning models on the test set.**

| Model | Popularity task | | | | Danceability task | | | |
|---|---|---|---|---|---|---|---|---|
| | MAE | MSE | RMSE | R2 | MAE | MSE | RMSE | R2 |
| MLP | 0.1577 | 0.0395 | 0.1989 | 0.1797 | 0.0998 | 0.0153 | 0.1239 | 0.3846 |
| DCN | 0.1560 | 0.0388 | 0.1970 | 0.1950 | 0.0900 | 0.0127 | 0.1128 | 0.4900 |
| TabNet | 0.1548 | 0.0383 | 0.1958 | 0.2049 | 0.0854 | 0.0116 | 0.1077 | 0.5348 |
| TabTransformer | 0.1541 | 0.0379 | 0.1948 | 0.2130 | 0.0826 | 0.0110 | 0.1049 | 0.5585 |
| FT-Transformer | 0.1428 | 0.0363 | 0.1832 | 0.2450 | 0.0701 | 0.0704 | 0.1001 | 0.5980 |
| GNM | 0.1530 | 0.0375 | 0.1937 | 0.2219 | 0.0808 | 0.0106 | 0.1031 | 0.5737 |
| Ours | **0.1228** | **0.0297** | **0.1723** | **0.3842** | **0.0667** | **0.0089** | **0.0942** | **0.6443** |

**Note:**
Bold denotes the best-performing value for each metric.

The experimental findings validate the effectiveness of our approach, demonstrating a significant improvement over current deep learning models for tabular data. The consistent reduction in prediction errors and the notable gains in explanatory power position our model as a promising solution for music-related predictive tasks. Future work may explore adaptations of this architecture for other music attributes or expand its use to real-world recommendation systems.

## LIMITATION AND FUTURE WORK

Despite the effectiveness of our proposed deep learning model in song rating and danceability prediction, certain limitations remain. One key limitation is the dependency on the quality and diversity of the training dataset. If the dataset does not comprehensively cover variations in song attributes, musical genres, and user preferences, the model may struggle with generalization to unseen data. Another limitation arises from the challenge of capturing subtle feature interactions. While the BiLSTM and Residual Network blocks help extract relevant information, the model may still miss intricate relationships between categorical and numerical features that impact song ratings. Enhancements such as more sophisticated fusion techniques or additional feature engineering could further improve performance.

Future research can enhance the joint modeling of danceability and song ratings by exploring contrastive and modality-disentangled learning techniques, as illustrated by *Lin et al. (2025)* and the multi-view approach described by *Zhang et al. (2024)*. Integrating user-centric contextual information, such as spatiotemporal usage patterns and physiological signals, could also benefit model personalization, drawing on methods from *Li et al. (2023)* and *Shen et al. (2022)*.

Furthermore, applying hybrid and ensemble learning strategies—such as the stacking of deep and classical models (*Nguyen et al., 2021*) may allow for more robust prediction by leveraging complementary feature sets. In parallel, recent advances in recommendation systems using graph neural networks, including the works of *Zuo et al. (2024)* and *Chen et al. (2025)*, could inform new ways of modeling complex user-song relationships and social interactions within music platforms.

In addition, the use of deep attention mechanisms, as successfully applied in music genre classification by *Nguyen et al. (2019)*, along with cross-attention frameworks for semi-supervised learning explored by *Liu et al. (2025)*, suggests opportunities for future models to dynamically focus on salient temporal and content-based features. Approaches from sequence data prediction (*Nguyen-Vo et al., 2021*, *2024*) may also inspire more sophisticated multimodal architectures capable of integrating diverse sources of musical and contextual data.

## CONCLUSIONS

This article presents a hybrid neural network framework that integrates a BiLSTM-based module for processing categorical features and a ResNet-based architecture for learning from numerical inputs, targeting both popularity prediction and danceability estimation tasks in the music domain. The proposed model consistently outperforms existing deep learning methods designed for tabular data, achieving superior predictive performance across all evaluation metrics. This improvement can be attributed to the model's ability to capture sequential and contextual dependencies in categorical features through bidirectional recurrent learning, while leveraging residual connections to extract deep and stable representations from numerical data. The integration of these components, combined with a cross attention mechanism, enables more effective modeling of interactions between heterogeneous feature types. As a result, the model demonstrates strong generalization capabilities and robustness in both prediction tasks.

### Funding
The authors received no funding for this work.

### Competing Interests
The authors declare that they have no competing interests.

### Author Contributions
- Wei Wu conceived and designed the experiments, performed the experiments, analyzed the data, performed the computation work, prepared figures and/or tables, authored or reviewed drafts of the article, and approved the final draft.

### Data Availability
The dataset used in this study is available at Kaggle: https://www.kaggle.com/datasets/edalrami/19000-spotify-songs.

The code is available in the Supplemental File.

## Supplemental Information

Supplemental information for this article can be found online at http://dx.doi.org/10.7717/peerj-cs.3009#supplemental-information.

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
