# Peer review of "Predicting danceability and song ratings using deep learning and auditory features"

_PeerJ Computer Science, doi:10.7717/peerj-cs.3009_

## Round 0.1 · original submission · Major Revisions

Please address the comments from the two reviewers, especially the suggestion to compare the proposed method with TabTransformer and FT-Transformer, and revise the manuscript accordingly.

Reviewer 1 ·

Basic reporting

The manuscript presents a deep learning model designed to predict how popular a song might become and how suitable it is for dancing. It is is well-organized, following a clear and logical structure that aligns with standard academic formats. Each section flows smoothly into the next, making the content easy to follow. The diagram illustrating the model's architecture is straightforward and easy to understand.
However, I have several comments to enhance the manuscript, as in the following sections.

Experimental design

- Preprocessing data, evaluation methods, assessment metrics, and model selection are adequately described.
- This paper provided detailed and methods described with sufficient information to replicate (code, dataset,computing infrastructure, reproduction script).
- To enhance the evaluation, the authors should compare their model with TabTransformer, which is specifically designed for handling categorical features using attention mechanisms.

Validity of the findings

- The current ablation study offers valuable insights into the model's components. However, it would be beneficial to include an experiment where the BiLSTM component is replaced with a Transformer-based module. This comparison would provide a clearer understanding of why BiLSTM was chosen over Transformer architectures for temporal modeling.
- The conclusion is well-written and summarizes the study effectively. Nonetheless, discussing the results of the suggested Transformer-based experiment would add depth to the analysis and offer meaningful directions for future research.

Additional comments

Please add citations for GRU in the relevant sections.

Reviewer 2 ·

Basic reporting

The manuscript is well-written and generally follows academic standards. The introduction clearly explains the motivation behind the study. However, it would benefit from a dedicated discussion on "danceability" as a musical concept. Including a definition and exploring its significance in previous research would provide better context for the second task of the study. This addition would help readers understand why predicting danceability is important and how it has been approached in past studies.

Experimental design

The experimental setup shows potential, but some details are missing:
+ It is unclear how the dataset was split into training, validation, and testing sets, and how many samples exist in each. A table summarizing the data distribution would be helpful.
+ Details on the training strategy (e.g., optimizer, learning rate, scheduling policy ...) should be provided for clarity and reproducibility.
+ The authors should consider including model configuration parameters (number of layers, hidden dimensions, dropout rate, etc.). This information is essential for replication and evaluation of architectural choices.
+ While the paper compares the model to traditional baselines and some deep learning approaches, the authors may want to compare the proposed model with FT-Transformer, which has shown strong performance on tabular data. Including this baseline would strengthen the evaluation. Another option is to compare the model with TabTransformer, another advanced deep learning architecture for tabular data.

Validity of the findings

+ Conclusions are well stated and limited to supporting results.
+ The experiments and evaluations are performed satisfactorily.
+ While not strictly necessary, including the formulas for the evaluation metrics could enhance the manuscript's clarity.

---

## Round 0.2 · accepted · Accept

All of the reviewers' comments have been addressed. The revised manuscript meets the standard of the journal. It can be accepted for publication.

Reviewer 1 ·

Basic reporting

The revised manuscript now fully meets the requirements for clear, unambiguous, and professional English, provides a well-contextualized and relevant introduction and literature review, adheres to PeerJ’s structural and disciplinary standards.

Experimental design

Methods described with sufficient detail & information to replicate.
The evaluation methods, assessment metrics, and model selection methods are adequately described.
The revised manuscript now includes a comparative evaluation of our model against TabTransformer, which is specifically designed for handling categorical features with attention mechanisms, as recommended.

Validity of the findings

The experiments and evaluations were performed satisfactorily. No further comments.

Additional comments

The author has addressed the reviewers' recommendations and made substantial revisions to the manuscript. I recommend that it be accepted.

Reviewer 2 ·

Basic reporting

The revised version is well-organized and clearly written. I don't have any other comments.

Experimental design

I have no further comments.

Validity of the findings

I have no further comments.

Additional comments

No further comments.